Biogeographic patterns in the cartilaginous fauna (Pisces: Elasmobranchii and Holocephali) in the southeast Pacific Ocean

Bustamante Carlos 1 2 c.bustamantediaz@uq.edu.au
Vargas-Caro Carolina 1 2
Bennett Michael B. 1
1 School of Biomedical Sciences, The University of Queensland , St. Lucia, Queensland , Australia
2 Programa de Conservación de Tiburones (Chile) , Valdivia , Chile
Stewart Gavin
Electronic publication date: 2014 May 29
Publication date: 2014
Volume: 2
Electronic Location ID: e416
Received 2014 Mar 26; Accepted 2014 May 15
Copyright: © 2014 Bustamante et al.
Copyright year: 2014
Copyright holder: Bustamante et al.
License: This is an open access article distributed under the terms of the Creative Commons Attribution License, which permits unrestricted use, distribution, reproduction and adaptation in any medium and for any purpose provided that it is properly attributed. For attribution, the original author(s), title, publication source (PeerJ) and either DOI or URL of the article must be cited.
License URL: https://creativecommons.org/licenses/by/4.0/

Keywords: Shark, Chimaera, Skate, Diversity, Trawling, CPUE, Chile, Chondrichthyes

Funding: Fondo de Investigacion Pesquera (FIP) No. 2005-61 Caracterización del fondo marino entre la III y X regiones CONICYT–Becas Chile and TUAP–Graduate School of The University of Queensland The study was funded by Fondo de Investigación Pesquera (FIP No. 2005-61): “Caracterización del fondo marino entre la III y X regiones”. CB and CV were supported by CONICYT–Becas Chile and TUAP–Graduate School of The University of Queensland. The funders had no role in study design, data collection and analysis, decision to publish, or preparation of the manuscript.

==============================
The abundance and species richness of the cartilaginous fish community of the continental shelf and slope off central Chile is described, based on fishery-independent trawl tows made in 2006 and 2007. A total of 194,705 specimens comprising 20 species (9 sharks, 10 skates, 1 chimaera) were caught at depths of 100–500 m along a 1,000 km transect between 29.5°S and 39°S. Sample site locations were grouped to represent eight geographical zones within this latitudinal range. Species richness fluctuated from 1 to 6 species per zone. There was no significant latitudinal trend for sharks, but skates showed an increased species richness with latitude. Standardised catch per unit effort (CPUE) increased with increasing depth for sharks, but not for skates, but the observed trend for increasing CPUE with latitude was not significant for either sharks or skates. A change in community composition occurred along the depth gradient with the skates, Psammobatis rudis, Zearaja chilensis and Dipturus trachyderma dominating communities between 100 and 300 m, but small-sized, deep-water dogfishes, such as Centroscyllium spp. dominated the catch between 300 and 500 m. Cluster and ordination analysis identified one widespread assemblage, grouping 58% of sites, and three shallow-water assemblages. Assemblages with low diversity (coldspots) coincided with highly productive fishing grounds for demersal crustaceans and bony fishes. The community distribution suggested that the differences between assemblages may be due to compensatory changes in mesopredator species abundance, as a consequence of continuous and unselective species removal. Distribution patterns and the quantitative assessment of sharks, skates and chimaeras presented here complement extant biogeographic knowledge and further the understanding of deep-water ecosystem dynamics in relation to fishing activity in the south-east Pacific Ocean.

Introduction

Cartilaginous fishes play an important role as top predators and have complex distribution patterns (Wetherbee & Cortés, 2004), affecting the structure and function of marine communities through interactions with other trophic links in food webs to which they belong (Ferretti et al., 2010). Spatial distribution patterns of marine fishes in the south-east Pacific Ocean are poorly understood, and most studies of demersal communities have focused on the ecology of continental shelf fauna at depths of between 20 and 150 m (Brattström & Johanssen, 1983; Ojeda, 1983; Carrasco, 1997; Ojeda, Labra & Muñoz, 2000; Camus, 2001; Sellanes et al., 2007). Descriptions of geographical patterns of marine fishes have been restricted to littoral species (Mann, 1954; Pequeño, Rucabado & Lloris, 1990), and based on regional inventories (Ojeda, Labra & Muñoz, 2000). A general lack of quantification of species abundance limits our understanding of the functional biodiversity of the continental shelf of Chile (Pequeño, 1989; Bustamante, Vargas-Caro & Bennett, in press).

Chile has a cartilaginous fish fauna that is relatively rich when compared with warm-temperate countries in South America (Bustamante, Vargas-Caro & Bennett, in press), but poor in the global context despite having one of the largest maritime territories in the world (Cubillos, 2005). Species checklists and biological observations constitute the first approaches in the study of the cartilaginous fish fauna in the Chilean marine ecosystem and there are a number of studies that have reported on elasmobranch species around the central and southern continental shelf, from both fishery-dependent and -independent surveys (Meléndez & Meneses, 1989; Pequeño, 1989; Pequeño, Rucabado & Lloris, 1990; Pequeño & Lamilla, 1993). In northern Chile, bycatch analysis of the crustacean trawl fishery has contributed to knowledge of the continental slope ecosystem through the description of biological diversity, composition and structure of the demersal fish fauna over a wide depth range (Sielfeld & Vargas, 1999; Acuña et al., 2005; Menares & Sepúlveda, 2005). While fishery-dependent studies offer a description of diversity and species assemblages of cartilaginous fishes, using catch per unit effort (CPUE) as a proxy for abundance (Acuña et al., 2005), they generally lack the ability to adequately identify or provide quantitative information on species richness, abundance hotspots and conspecific assemblages that are required for a better understanding of marine ecosystem interrelationships (Kyne & Simpfendorfer, 2007).

The aim of the present study is to analyse abundance and species richness of cartilaginous fishes of the continental shelf and slope in Chile to identify patterns in the geographical and bathymetric distribution of sharks, skates and chimaeras in the south-east Pacific Ocean to complement existing biogeographic models, and improve the understanding of deep-water ecosystem dynamics in the context of fishing activities.

Material and Methods

Data were collected through direct observation of total catch on fishery-independent surveys made along the Chilean continental slope and shelf as part of a broader project to assess the biological and oceanographical characteristics of the Chilean seafloor (Melo et al., 2007). Surveys were carried out on-board two fishing vessels, “Crusoe I” and “Lonquimay”, equipped as oceanographic research platforms. Fishing gear comprised a bottom trawl constructed from 3 mm diameter polyamide nylon with 50 mm stretch-measured diamond-mesh in the tunnel and cod-end. The trawl had a 24 m headrope, a 28 m footrope, and a stretched circumference of 34 m with an average net opening during tows of 11 m. Tows lasted 18–53 min at a speed of 3.7 km h−1 which resulted in a swept area of 12.2–35.9 km2. Geometric construction of fishing gear and tow speed were used to calculate CPUE which was standardised as individuals per hour and square kilometre swept (ind km−2 h−1). For each species, CPUE data were calculated separately and log-transformed (Log (CPUE + 1)) in order to assess the departure of original data from normality. Geographic coordinates and depth of each trawl were recorded for each tow.

A total of 128 tows were made in 32 sites grouped in eight regions, numbered from north to south as zones 1 to 8, that span approximately 1,000 km between the latitudes 29.5°S and 39°S (Fig. 1). Survey data were collected from sites in four depth strata (labelled as A: 100 and 199 m, B: 200–299 m, C: 300–399 m and D: 400–499 m) with four pseudoreplica tows in each site (16 tows per zone with 4 tows per site). Zones 1, 2, 4 and 5 were sampled in July/August 2006, zones 6, 7 and 8 in November/December 2006, and zone 3 was sampled twice, in July 2006 and again in March 2007. Each site was recorded and coded with the zone (1 to 8), depth strata (A to D) and pseudoreplica tow (numbered 1 to 4), i.e., tow coded as “1.A.2” represents the second tow made in zone 1, between 100 and 199 m depth.

Figure 1 Study area.

Map of (A) Chile showing location of study area (inset box) and (B) location of zones (Z1 to Z8) and sampling sites (circles). Commercial trawl intensity is indicated in (B), in terms of tows per nautical mile (nmi). Modified after Melo et al. (2007).

This study was carried out in accordance with the “standards for the use of animals in research” approved by the Animal Care and Ethics Committee of the Universidad Austral de Chile (UACH/FIP 2005-61). Capture of fishes during this study was permitted through Fisheries Undersecretariat Research Permit Number 1959-06, 2931-06 and 181-07 issued by Ministry of Economy, Development and Tourism.

Community definition

All cartilaginous fishes captured during surveys were counted and identified to species. A number of individuals caught (∼1%) were landed frozen to validate on-board identification using diagnostic features described in literature (Compagno, 1984a; Compagno, 1984b; Lamilla & Sáez, 2003; Lamilla & Bustamante, 2005; Ebert, Fowler & Compagno, 2013). Species diversity was calculated from the number of species at each tow; and compared using the Shannon diversity index (H according to Spellerberg & Fedor, 2003) by depth and zone.

Species richness (S) was calculated per depth stratum in each zone, and is defined as the number of species within a specific number of individuals sampled (Kempton, 1979). Relative frequency of occurrence (FO) was determined for each species to explore the variability of species’ occurrence along bathymetric and latitudinal gradients; and is expressed as a percentage of occurrence of a species in relation to the total number of tows within sites and zones. Three categories of FO were determined according to Solervicens (1973): regular species, where FO =   >50%; accessory species, where FO = 25–49% and; incidental species, where FO = 10–24%. Latitudinal and bathymetric gradients of species diversity of the major taxonomic groups (sharks and skates) were compared using analysis of covariance (ANCOVA) with significance accepted at P < 0.05.

Community structure

Faunal assemblages and geographic patterns of cartilaginous fishes were determined through a global similarity matrix. Species composition and abundance in each tow were considered for the entire study area with CPUE values fourth-root transformed to balance outliers (rare and abundant species). Sampling sites were sorted by an agglomerative hierarchical cluster and through non-dimensional metric scaling (nMDS) considering the global similarity matrix (Clarke, 1993; Clarke & Warwick, 1994). Log-transformed CPUE values were used for hierarchical agglomerative clustering with group-averaging linking, based on the Bray–Curtis similarity measure to delineate groupings with a distinct community structure. A one-way ANOSIM was used to establish possible differences between sampling site groups. Additionally, a SIMPER analysis was used to determine the contribution of each species to the average Bray–Curtis dissimilarity between groups. All indices and statistical procedures were made using software PRIMER v.6.0 (Plymouth Marine Lab, Plymouth, UK).

Results

From 32 sites sampled, the total catch was 194,705 cartilaginous fishes from the 76 towsthat contained specimens, of which 2,725 individuals were landed and examined. In 52 tows (40.6% of the total) there was no catch of cartilaginous fishes and were thus excluded from the remaining analysis. A total of 20 species (nine sharks, ten skates and one chimaera) was confirmed (Table 1). Note, that for the purpose of the current study the term ‘skate’ includes Torpedo tremens. Bathymetrically, the shallowest depth stratum (100–199 m) and latitudinally, the northernmost zone (zone 1) yielded the lowest percentage occurrence of cartilaginous fishes caught in 3.13% and 37.5% of tows respectively (Table 2). The greatest number of species caught per family was five, in the family Arhynchobatidae, followed by the families Rajidae (four species), Etmopteridae and Scyliorhinidae (both three species). The Hexanchidae, Somniosidae, Centrophoridae, Torpedinidae and Chimaeridae were each represented by a single species (Table 1).

Table 1 Taxonomic composition of samples analysed.

Depth and latitudinal range of cartilaginous fishes caught during surveys.

Order	Family	Species	Depth range (m)	Latitudinal range (°S)	
Hexanchiformes	Hexanchidae	Hexanchus griseus (Bonnaterre 1788)	358–376	35–35.1	
Squaliformes	Etmopteridae	Aculeola nigra de Buen 1959	262–492	29.4–36.5	
Squaliformes	Somniosidae	Centroscymnus macracanthus Regan 1906	455	33.3	
Squaliformes	Etmopteridae	Centroscyllium granulatum Günther 1887	262–482	33.2–38.9	
Squaliformes	Etmopteridae	Centroscyllium nigrum Garman 1899	335–455	32–38.8	
Squaliformes	Centrophoridae	Deania calcea (Lowe 1839)	362–492	29.5–38.9	
Carcharhiniformes	Scyliorhinidae	Apristurus brunneus (Gilbert 1892)	443–461	34.5–36.5	
Carcharhiniformes	Scyliorhinidae	Apristurus nasutus de Buen 1959	338–482	29.5–38.9	
Carcharhiniformes	Scyliorhinidae	Bythaelurus canescens (Günther 1878)	237–492	29.4–38.9	
Rajiformes	Arhynchobatidae	Bathyraja albomaculata (Norman 1937)	356–436	37.8–38.7	
Rajiformes	Arhynchobatidae	Bathyraja brachyurops (Fowler 1910)	482	38.9	
Rajiformes	Arhynchobatidae	Bathyraja multispinis (Norman 1937)	445	36.4	
Rajiformes	Arhynchobatidae	Bathyraja peruana McEachran & Miyake 1984	243–492	29.6–38.9	
Rajiformes	Arhynchobatidae	Psammobatis rudis Günther 1870	240–475	32–38.8	
Rajiformes	Rajidae	Gurgesiella furvescens de Buen 1959	362–484	29.4–32	
Rajiformes	Rajidae	Zearaja chilensis (Guichenot 1848)	159–476	33.3–38.7	
Rajiformes	Rajidae	Dipturus trachyderma (Krefft & Stehmann 1975)	234–482	32–38.9	
Rajiformes	Rajidae	Rajella sadowskii (Krefft & Stehmann 1974)	475	33.4	
Rajiformes	Torpedinidae	Torpedo tremens de Buen 1959	149–376	34.5–38.9	
Chimaeriformes	Chimaeridae	Hydrolagus macrophthalmus de Buen 1959	430–483	29.6–37.8	

Table 2 Summary of the sampling design.

Percentage of tows with cartilaginous fishes in the catch, species richness (S) and total number (N) of cartilaginous fishes caught in each zone and depth stratum.

Zone	Catch (%)	S	N	
1	37.5	7	2,921	
2	56.25	10	14,871	
3	56.25	11	12,199	
4	62.5	11	15,058	
5	68.75	10	23,224	
6	56.25	12	60,651	
7	75	12	47,862	
8	62.5	12	17,919	
Depth stratum (m)	Catch (%)	S	N	
100–200	3.13	2	203	
200–300	65.63	8	18,907	
300–400	78.13	14	58,597	
400–500	90.63	18	116,998	

Community definition

Species richness fluctuated between one and six species per site with no significant differences between sharks and skates in slopes of the regression (ANCOVA; F = 0.826; df = 1, 117; P = 0.365; Fig. 2), but there were significant differences in the intercepts (ANCOVA; F = 24.972; df = 1, 117; P > 0.001). There was no significant relationship between species richness and latitude for sharks, but species richness for skates increased with increasing latitude (Figs. 2A and 2C). Chimaeras were absent in the catch from zones 6 and 8, but occurred in the other six zones (Fig. 2E). Species richness increased significantly with depth for sharks, but not for skates (Figs. 2B and 2D). The slopes and intercepts of the regressions were significantly different (ANCOVA, F = 17.06; df = 1, 117; P > 0.001 and F = 13.954; df = 1, 117; P > 0.001, respectively). Chimaeras were restricted to 430–480 m within the deepest depth stratum, and were observed off most of the central coast of Chile, between approximately 29.5° and 37.5°S (Figs. 2E and 2F).

Figure 2 Variation of species richness in cartilaginous fishes.

Latitudinal and bathymetric changes of species richness of sharks (A–B), skates (C–D) and chimaeras (E–F) across the study area. Fitted least-square regression model (solid line) and statistical significance are indicated in each case.

The CPUE per site ranged widely, from 5.5 to 2,785 ind km−2 h−1 among individual sites and 728 to 7,942 ind km−2 h−1 among zones (Table 3). Log-transformed CPUE increased with latitude for both sharks and skates, although the slopes of the regressions were not significantly different (Figs. 3A and 3C). Based on latitude, the ANCOVA did not reveal significant differences in slope (F = 0.412; df = 1, 117; P = 0.523), but did in elevation between sharks and skates (F = 43.942; df = 1, 117; P > 0.001). There was a significant effect of depth on the CPUE for sharks, but not for skates (Figs. 3B and 3D), and there was a significant difference between the slopes and elevations of the regressions (ANCOVA; F = 19.59; df = 1, 117; P > 0.001; F = 31.12; df = 1, 117; P > 0.001, respectively). For chimeras, the CPUE was generally low across the species’ latitudinal range (Fig. 3E).

Figure 3 Variation in abundance of cartilaginous fishes in Chile.

Latitudinal and bathymetric changes of relative abundance (Log (CPUE + 1)) of sharks (A–B), skates (C–D) and chimaeras (E–F) across the study area. Fitted least-square regression model (solid line) and statistical significance are indicated in each case.

Table 3 Catch per unit effort of shark, skates and chimaeras per geographic zone.

Abundance, as total CPUE (ind km−2 h−1) of cartilaginous fishes caught during surveys in each zone (geographic location of zones is indicated in Fig. 1).

Zone	
Species	1	2	3	4	5	6	7	8	
H. griseus	—	—	—	—	54.7	—	—	—	
A. nigra	130	249.4	208	390	10	11	—	—	
C. macracanthus	—	—	9.2	—	—	—	—	—	
C. granulatum	—	770.6	109.9	259.8	64.7	4,611	1,730	577.6	
C. nigrum	—	257.5	752.6	363.8	2,845.1	1,639.8	435.7	5.2	
D. calcea	15	54.7	68.5	37.8	41.5	28.4	122.1	85.1	
A. brunneus	15	—	—	—	—	15.5	326.3	206.7	
A. nasutus	—	—	—	30.6	—	59.2	—	—	
B. canescens	272.7	312.5	403.8	476.5	483.2	1084.4	361.4	160.5	
B. albomaculata	—	—	—	—	—	—	14.5	5	
B. brachyurops	—	—	—	—	—	—	—	4.7	
B. multispinis	—	—	—	—	—	8.4	—	—	
B. multispinis	42.4	52	65.7	121.8	21.5	50.2	29	92	
P. rudis	—	32.7	71.0	38.5	192.2	77.1	154.2	14.9	
G. furvescens	239.5	55.5	—	—	—	—	—	—	
Z. chilensis	—	—	9.2	—	—	21	984.1	5	
D. trachyderma	—	55.8	—	127.8	159.3	336.2	100.6	395.3	
R. sadowskii	—	—	38.2	—	—	—	—	—	
T. tremens	—	—	—	18.7	—	—	10.1	4.4	
H. macrophthalmus	14.2	17.6	9.2	15.2	63.9	—	5.9	—	
Total	728.8	1,858.3	1,745.3	1,880.5	3,936.1	7,942.2	4,273.9	1,556.4	

Diversity index (H) was not influenced by latitude for sharks, but increased significantly for skates (Fig. 4; ANCOVA; F = 5.056; df = 1, 117; P = 0.263) and the intercepts were significantly different (ANCOVA; F = 15.92; df = 1, 117; P > 0.0001). Values of H for sharks averaged approximately 0.6 across the eight zones, but showed high variability among sites in each zone (Fig. 4A). For skates, there were zero-values for H in all zones, particularly zone 1, but values of up to approximately 1.1 also occurred at sites in the central and southern zones (Fig. 4C). Significant differences were observed in the slopes and intercepts of the regression between sharks and skates based on depth (ANCOVA; F = 15.35; df = 1, 117; P > 0.001 and F = 8.40; df = 1, 117; P > 0.001). Diversity index for sharks was markedly higher in waters over about 325 m deep, and was almost absent in shallowed depth strata (Fig. 4B). Skate diversity varied considerably within most depth strata and, overall, showed no significant trend with depth (Fig. 4D).

Figure 4 Variation in diversity of cartilaginous fishes in Chile.

Latitudinal and bathymetric changes of Shannon diversity index (H) of sharks (A–B) and skates (C–D) across the study area. Fitted least-square regression model (solid line) and statistical significance are indicated in each case.

Three incidental species (Bathyraja multispinis, Dipturus trachyderma, Torpedo tremens) and two regular species (Psammobatis rudis, Zearaja chilensis), represent the community at 200–299 m depth. Hexanchus griseous and T. tremens are regular species, along with six accessory species in the 300–399 m depth stratum. Hexanchus griseus was restricted to this stratum, whereas T. tremens was also captured at shallower depths. Centroscymnus macracanthus, Apristurus nasutus, Bathyraja peruana, Bathyraja albomaculata, Rajella sadowskii and Hydrolagus macrophthalmus were only found in the deepest stratum (400–499 m), whereas there were nine other regular species that were also represented in shallower strata (Table 5).

A taxonomic change in community composition occurred along the depth gradient. Three skates, Psammobatis rudis, Zearaja chilensis and Dipturus trachyderma dominated communities between 100 and 300 m accounting for >80% of total cartilaginous fish CPUE, but as depth increased there was a major shift in community, as small-sized, deep-water dogfishes, such as Centroscyllium spp. came to dominate the catch (Fig. 3, Table 4). Other contributors to this species-complex change were relative reductions in Bythaelurus canescens and small-sized skates (i.e., Psammobatis rudis and Gurgesiella furvescens) (Tables 4 and 5).

Table 4 Occurrence of shark, skates and chimaeras per geographic zone.

Frequency of occurrence of cartilaginous fishes caught during surveys in each zone (geographic location of zones is indicated in Fig. 1).

Species	Zone	
	1	2	3	4	5	6	7	8	
H. griseus	—	—	—	—	100	—	—	—	
A. nigra	7.2	27.6	20.1	43.2	0.8	1.2	—	—	
C. macracanthus	—	—	100	—	—	—	—	—	
C. granulatum	—	8.7	1.1	2.9	0.5	49.6	27.2	10	
C. nigrum	—	4.6	11.8	6.5	37.7	28.2	10.9	0.1	
D. calcea	1.4	10.5	11.5	7.3	5.9	5.9	32.8	25.2	
A. brunneus	0.9	—	—	—	—	1.9	57.3	39.9	
A. nasutus	—	—	—	35.2	—	64.8	—	—	
B. canescens	4	9.1	10.3	13.9	10.4	30.3	14.7	7.2	
B. albomaculata	—	—	—	—	—	—	72.6	27.4	
B. brachyurops	—	—	—	—	—	—	—	100	
B. multispinis	—	—	—	—	—	100	—	—	
B. peruana	4.3	10.4	11.5	24.5	3.2	9.6	8.1	28.4	
P. rudis	—	5.6	10.6	6.6	24.2	12.5	36.7	3.9	
G. furvescens	68.4	31.6	—	—	—	—	—	—	
Z. chilensis	—	—	0.6	—	—	1.4	97.5	0.5	
D. trachyderma	—	94.5	—	10.4	9.5	26.1	12.6	36.9	
R. sadowskii	—	—	100	—	—	—	—	—	
T. tremens	—	—	—	18.7	—	—	35.6	17	
H. macrophthalmus	6.9	17	7.7	14.8	45.6	—	8	—	

Table 5 Catch per unit effort and occurrence of shark, skates and chimaeras per depth strata sampled.

Abundance, as total CPUE (ind km−2 h−1) and frequency of occurrence (FO) of cartilaginous fishes caught in each depth stratuma.

Species	CPUE	FO	
	Depth stratum	Depth stratum	
	A	B	C	D	A	B	C	D	
H. griseus	—	—	54.7	—	—	—	100	—	
A. nigra	—	4.1	45.8	948.1	—	0.4	4.6	95	
C. macracanthus	—	—	—	9.2	—	—	—	100	
C. granulatum	—	85.4	3,258.8	4,779.3	—	1.1	40.1	58.8	
C. nigrum	—	—	1,541.1	4,758.6	—	—	24.5	75.5	
D. calcea	—	—	220.9	232.3	—	—	48.7	51.3	
A. brunneus	—	—	23.2	540.2	—	—	4.1	95.9	
A. nasutus	—	—	—	89.8	—	—	—	100	
B. canescens	—	18.7	1,121.4	2,415.4	—	0.5	31.6	67.9	
B. albomaculata	—	—	9.4	10.0	—	—	48.4	51.6	
B. brachyurops	—	—	—	4.7	—	—	—	100	
B. multispinis	—	—	—	8.4	—	—	—	100	
B. peruana	—	61.1	214.2	199.3	—	12.9	45.1	42	
P. rudis	—	430.1	122.4	28.1	—	74.1	21.1	4.8	
G. furvescens	—	—	38.4	254.3	—	—	13.1	86.9	
Z. chilensis	13.7	951.1	39.9	14.5	1.3	93.3	3.9	1.4	
D. trachyderma	—	375.4	431.2	278.5	—	34.6	39.7	25.7	
R. sadowskii	—	—	—	38.2	—	—	—	100	
T. tremens	5.3	6.5	21.4	—	16.1	19.6	64.3	—	
H. macrophthalmus	—	—	—	126.0	—	—	—	100	
Total	19.0	1,932.4	7,142.8	14,734.9					
Notes.

a Depth strata are A, 100–199 m; B, 200–299 m; C, 300-0399; D, 400–499.

Community structure

Agglomerative hierarchical cluster analysis (Fig. 5) revealed four major fish assemblages (I–IV) at similarity level of 40%, and one outlier. The ANOSIM showed that the four assemblages were significantly separated from each other (n = 76, R Global = 0.68; P > 0.01), with the outlier characterised by the presence of one single species (Bathyraja peruana) with the lowest total CPUE (8.6 ind km−2 h−1). Geographically, assemblage I grouped 11 sites located north of Coquimbo to Valparaíso (zones 1–3, Fig. 1) and between depths of 237 to 379 m, with an average of CPUE of 56.3 ind km−2 h−1 for 10 species (5 sharks and 5 skates). This community was dominated by Centroscyllium nigrum that comprised 34.3% of the CPUE, Bythaelurus canescens (22.2% CPUE) and Psammobatis rudis (11.5% CPUE) (Table 5). Assemblage II included the largest number of sites (45), taxa (20) and specimens (average CPUE = 475 ind km−2 h−1). Sites in this assemblage were scattered over the entire study area and occupied a depth range of 335–492 m. Prominent species in this assemblage were C. granulatum (37.6% CPUE), C. nigrum (28.5% CPUE), and B. canescens (15.9% CPUE) (Table 5). Assemblage III comprised 10 relatively shallow sites (149–262 m) in the most southerly zone offshore from Concepción, the second largest port in Chile. The skates Z. chilensis and D. trachyderma dominated this assemblage of 6 species with 83.3% of the assemblage CPUE (158 ind km−2 h−1; Table 5). Assemblage IV grouped 10 relatively shallow sites (243–281 m) located south of Valparaíso in zones 4, 5 and 6. This assemblage had the lowest diversity (5 species) and abundance (39.9 ind km−2 h−1). Two species, Psammobatis rudis and C. granulatum, were the most abundant species accounting for 63.4% and 20.4% of CPUE respectively (Table 6).

Figure 5 Cluster of assemblages.

Agglomerative hierarchical cluster indicating the clustering of the four assemblages. Site grouping is colour coded and indicates 40% similarity. Sites are coded following zone (1 to 8), depth strata (A to D) and pseudoreplica (1 to 4).

Table 6 SIMPER summary.

Average abundance (ind km−2 h−1) and percentage of contribution per species in each assemblage (n indicates the number of sites included per assemblage).

Species/Assemblage	I (n = 11)	II (n = 45)	III (n = 9)	IV (n = 10)	
	Avg.	%	Avg.	%	Avg.	%	Avg.	%	
H. griseus	—	—	1.2	0.3	—	—	—	—	
A. nigra	4.1	7.2	21.1	4.4	—	—	0.4	1.0	
C. macracanthus	—	—	0.2	0.0	—	—	—	—	
C. granulatum	—	—	178.5	37.6	1.1	0.7	8.1	20.4	
C. nigrum	19.3	34.3	135.3	28.5	—	—	—	—	
D. calcea	1.4	2.4	9.7	2.0	—	—	—	—	
A. brunneus	1.4	2.4	12.2	2.6	—	—	—	—	
A. nasutus	—	—	2.0	0.4	—	—	—	—	
B. canescens	12.5	22.2	75.9	16.0	—	—	—	—	
B. albomaculata	—	—	0.4	0.1	—	—	—	—	
B. brachyurops	—	—	0.1	0.0	—	—	—	—	
B. multispinis	—	—	0.2	0.0	—	—	—	—	
B. peruana	2.5	4.5	8.9	1.9	4.1	2.6	—	—	
P. rudis	6.5	11.5	2.0	0.4	18.3	11.6	25.3	63.4	
G. furvescens	2.8	4.9	5.9	1.2	—	—	—	—	
Z. chilensis	0.8	1.5	1.2	0.3	106.2	67.4	—	—	
D. trachyderma	5.1	9.0	16.3	3.4	26.8	17.0	5.7	14.2	
R. sadowskii	—	—	0.8	0.2	—	—	—	—	
T. tremens	—	—	0.5	0.1	1.1	0.7	—	—	
H. macrophthalmus	—	—	2.8	0.6	—	—	—	—	

Ordination analysis (nMDS) produced similar results to cluster analysis with four assemblages (Fig. 6). The outlier observed (zone 3, site B, tow 1) was a tow off Valparaíso apparently separated from other tows due to the presence of a single species (Bathyraja peruana) with low abundance (8.5 ind km−2 h−1). SIMPER analysis showed low average within-assemblage similarity of 29.9–38.6% for all assemblages. Two main consolidating species, P. rudis and D. trachyderma were identified within each assemblage, and accounted for 100% within-assemblage similarity in assemblage III; 59.4% in assemblage IV and >6% in assemblages I and II, respectively. Unlike within-assemblage similarity, the between-assemblage dissimilarity levels in all four assemblages were high, ranging from 92.7 to 96.7%. Psammobatis rudis, Bythaelurus canescens, Centroscyllium nigrum and Dipturus trachyderma, accounted for 80.7% of total (84.2%) dissimilarity between assemblages I and III. Nine species together contributed 92.9% towards total (96.7%) dissimilarity between assemblages I and II. Eight species were responsible for 91.9% (95.1%) and 90.5% (94.3%) of total dissimilarity in both, assemblages II and III and assemblages II and IV respectively. Finally, seven species contributed 92% towards total (93.4%) dissimilarity between assemblages II and III; while between assemblages III and IV, Zearaja chilensis, Dipturus trachyderma, Psammobatis rudis and Centroscyllium granulatum accounted for 91.9% of total (92.7%) dissimilarity.

Figure 6 nMDS of sites.

Ordination in two-dimensions using non-dimensional metric scaling indicating the clustering of the four assemblages. Sites grouping is colour coded and indicate 40% similarity. Colour and site codes follows Fig. 5.

Discussion

Trawling has long been used to explore waters off the central-north and central-south coasts of Chile in order to identify regions where benthic crustaceans and teleost fishes of commercial interest occur in high abundance (Sielfeld & Vargas, 1999; Menares & Sepúlveda, 2005). Currently, trawl-fishing effort is centred, but not restricted, on squat lobsters (Cervimunida johni and Pleuroncodes monodon), deep-water shrimps (Heretocarpus reedi), hakes (Merluccius gayi and M. australis) and Chilean horse mackerel (Trachurus murphyi). The abundance of these target species is estimated through regular trawl surveys to allow the fishing effort to be adjusted to achieve ‘maximum sustainable yield’. A useful by-product of such surveys has been the production of species checklists that have enriched knowledge of Chile’s national marine biodiversity (Pequeño, 2000; Acuña et al., 2005). These extensive fishery-dependent and independent surveys, that include cartilaginous fishes in the catch, are conducted annually in central Chilean waters (c. 21.5–38.5°S). For example, between 1994 and 2004, exploratory surveys for demersal crustaceans comprised 6,143 trawl hauls made at depths of 100–500 m (Acuña et al., 2005). Although 13 shark, 8 skate and 1 chimaera species were caught, published data are limited to a simple indication of the latitudinal range for each species (Acuña et al., 2005). The absence of quantitative data on the species’ abundance, particularly in respect of fishing effort, location (latitude) and depth provides a challenge for management, whether for exploitation or for conservation. It is also of relevance to note that these fishery-dependent and independent surveys report on the diversity of animals from areas that are subject to continuous and often intense fishing activity which is implicated in the decline in species richness (Wolff & Aroca, 1995).

There has also been a number of fishing-independent studies, such as Ojeda (1983), that reported the presence of 2 shark and 3 skate species from 118 hauls made at depths of over 500 m on a trawl survey in austral Chile (52°S–57°S). Further north, 133 hauls made between 31°S and 41°28′S at depths of 50–550 m produced 7 shark, 5 skate and 1 chimaera species (Menares & Sepúlveda, 2005). In central Chile, Meléndez & Meneses (1989) reported 11 shark species from 173 hauls in exploration surveys using bottom trawl nets between 18°S and 38°30′S and at depths of 500–1260 m. In the most northerly survey, between 18°S and 21°S, the same gear type used over a wider depth range (30–1050 m) resulted in 4 shark, 4 skate and 1 chimaera species from 21 hauls (Sielfeld & Vargas, 1999). Each of these studies, however, also lacked quantification of the catch and are therefore of limited value, beyond providing information on the presence (or apparent absence) of species within a geographic region.

Community definition

The species richness observed in the current study (20 species), is higher than those found in surveys conducted previously in the region (Ojeda, 1983; Meléndez & Meneses, 1989; Sielfeld & Vargas, 1999; Ojeda, Labra & Muñoz, 2000; Acuña & Villarroel, 2002; Acuña et al., 2005; Menares & Sepúlveda, 2005). Variation in the reported species richness of cartilaginous fishes within the region among years may reflect the different gear types used, different effort, different depths sampled, and species misidentifications (Pequeño & Lamilla, 1993; Lamilla et al., 2010). While the species richness reported here is similar to that reported by Acuña et al. (2005), the cartilaginous fish community appears to differ between the two studies. Direct comparisons are somewhat speculative as while our study provides quantification of the fauna in terms of CPUE and FO while the results of Acuña et al. (2005) are limited to whether a species was present, in unreported abundance. Nevertheless, a couple of thematic differences are apparent with small, shallow-water skates (i.e., Psammobatis scobina, Sympterygia lima, S. brevicaudata and Discopyge tschudii) absent in our study, while deep-sea skates of the genera Bathyraja and Rajella were not caught in the earlier study (Fig. 7). These results suggest that, in comparison to our study, (a) shallower waters may have been sampled, and (b) the fishing effort in deeper waters was more limited in the study reported by Acuña et al. (2005). Taken together, these two studies indicate that at least 30 cartilaginous fishes inhabit (or did inhabit) the continental shelf and slope off central Chile; although some species showed pronounced latitudinal variation in distribution (e.g., Aculeola nigra, Centroscyllium nigrum, Gurgesiella furvescens) while in some others, the latitudinal extension is not reported (i.e., Bathyraja peruana, Sympterygia brevicaudata, S. lima, Discopyge tschudii).

Figure 7 Diagram of abundance and latitudinal range of cartilaginous fishes in Chile.

Latitudinal distribution and abundance (Log (CPUE + 1)) of cartilaginous fishes present in the continental shelf and slope of Chile. Solid lines represent species range reported by Acuña et al. (2005).

Species abundance was highly variable between zones with the lowest abundance in the north (zone 1). The abundance in the central and the most southern zones (2, 3, 4 and 8) was about double this value, in zones 5 and 7 it was four times as large and in zone 6 it was an order of magnitude greater. Interestingly, five species (e.g., Hexanchus griseus, Centroscymnus macracanthus, Bathyraja brachyurops, B. multispinis, Rajella sadowskii) were caught, mostly in low numbers, only within a single zone and within a single depth stratum. The pattern of occurrence suggests that the species are naturally uncommon or, more likely, that the trawl regime only sampled the upper end of their natural range (Fig. 7). In contrast, two species (e.g., Apristurus nasutus and Hydrolagus macrophthalmus) showed a marked preference for a particular depth stratum but occurred in more than one zone. Others species showed an obvious latitudinal variation in abundance, for example, Aculeola nigra was common in the north (zones 1–4), rare in central zones (5–6) and absent in the southern zones (7–8); whereas, Psammobatis rudis and Dipturus trachyderma showed the opposite trend. Both Centroscyllium species (C. granulatum and C. nigrum) have a high abundance in central Chile and are less common in both north and south, and appear to become extremely abundant with increasing depth. Between 300 and 500 m, the diversity further doubled and the abundance of most species increased. With the exception of two species (Zearaja chilensis and Torpedo tremens), all cartilaginous fishes were caught at depths below 200 m and most increase their abundance with depth. This relative absence of cartilaginous fishes in shallow waters (100–199 m) was both unexpected and difficult to explain, and needs to be addressed in future studies.

Community structure

Species richness of cartilaginous fishes in the south-east Pacific has been described to increase towards lower latitudes following the same geographic pattern of other marine fishes (Meléndez & Meneses, 1989; Pequeño, Rucabado & Lloris, 1990; Rohde, 1992; Pequeño & Lamilla, 1993; Camus, 2001); however, these observations are based on species inventories without reference to latitudinal or bathymetric ranges which obviously can have a marked influence on species distributions. Also, elasmobranch diversity in the Atlantic and Pacific oceans have been described to decrease with depth, especially below 1,000 m depth (Pakhomov et al., 2006; Priede et al., 2006). Our results provided evidence of an overall increase in species richness with increasing latitude and depth down to 500 m, in contrast to a decrease in diversity with increasing latitude demonstrated by littoral fishes (Ojeda, Labra & Muñoz, 2000), but similar to diversity gradients of benthic invertebrates and in the Northern Hemisphere described by Rex, Stuart & Coyne (2000). In our study, the latitudinal and bathymetric stability of assemblage II (Fig. 5), is consistent with a “transition intermediate area” as described by Camus (2001), and suggests that differences between assemblages were due to compensatory changes in mesopredator abundance (Navia et al., 2011). There is a correlation between the location of assemblages I, III and IV and intensive trawl fishing areas (Wolff & Aroca, 1995; Escribano, Fernandez & Aranis, 2003; Acuña et al., 2005). Continuous and unselective removal of certain species by commercial fisheries may explain in part the variation of species abundance among assemblages.

At the community level, the main assemblage (II) was distributed across the entire surveyed area comprising 58% of sites; and showed a high average dissimilarity to assemblages I, III and IV (96.7, 95.1 and 94.3% respectively). Differences were mainly due to the importance of small-sized sharks (Bythaelurus canescens, Centroscyllium granulatum and C. nigrum), although diversity of small-sized skates also contributed to overall dissimilarity. In our study assemblages I, III and IV represented ‘coldspots’ of diversity, similar to those found along the outer shelf in south-west Atlantic cartilaginous fish community (Lucifora et al., 2011). While those coldspots were simply defined as areas of low diversity, in the current study coldspots coincide with traditional fishing grounds. Commercial fisheries in Chile, in particular trawl-based activities, are likely to have a direct effect on cartilaginous fish community structure and distribution as has been previously documented for other marine fishes in central Chile (Arancibia & Neira, 2005).

Different levels of fishing pressure can generate multiple effects on the function of species and their interactions (Navia et al., 2011). High species richness and abundance represented in assemblage II, is consistent with a more stable community as high biodiversity has been linked to the stability of trophic networks through the complex interactions that arise among its components (Navia et al., 2011). In contrast, when there is an external disturbance, in this case differential exposure to fishing pressure, the result may be a complete reorganisation of the community (Bascompte, Melián & Sala, 2005).

Considering the overall species composition without counting rare species (defined in relation to low species abundance), such as Echinorhinus cookei and Centroscymnus owstonii, the absence of mid- to large-sized sharks is evident in our study (Fig. 7), however fishing gear selectivity and species catchability may influence the frequency of occurrence observed. Ferretti et al. (2010) described the ecological restructuring of demersal elasmobranch communities in fishing areas worldwide. Diversity and abundance of elasmobranchs erodes quickly as fisheries remove, unselectively, both small and large species despite the lower catchability of the latter. As large sharks disappear from the catch as fisheries develop, the community tends to become dominated by mesopredators. In the current study these mesopredators are predominantly small-sized sharks, which are more fecund and more resilient to fishing pressures than other elasmobranchs. Examples of similar community restructuring have been documented for trawl fishing areas in the Atlantic (Ellis et al., 2005), Gulf of Mexico (Shepherd & Myers, 2005), the Mediterranean Sea (Ferretti et al., 2008) and Australian waters (Graham, Andrew & Hodgson, 2001); although its extension to similar trawl fisheries elsewhere has not been properly evaluated due to a lack of temporal and seasonal catch-composition data for elasmobranch species.

Limitations and future directions

Previous research has identified two distinct biogeographic provinces based on multiple taxa along the Chilean coast, the Peruvian province in the north (4°–30°S) and the Magellanic province in the south (42°–56°S) (Camus, 2001). There is also an “intermediate area” between these two provinces that has been described as a rich, mixed-origin species’ transition zone for teleost fishes (Pequeño, 2000; Ojeda, Labra & Muñoz, 2000). Considering the limitations of geographic scale, the single main biogeographic province (assemblage II) that was identified between 29.5°S and 38.5°S only showed limited evidence of species more usually associated with the Peruvian and Magellanic provinces.

Fishery-independent surveys allowed us to explore an extensive area, including traditional commercial trawling zones and non-traditional fishing zones with similar effort. It should be mentioned that the methodology used was designed to sample demersal and bottom-dwelling species, and therefore the cartilaginous fish community’s definitions used here effectively excludes species that occur in mid- to surface waters and likely underestimates species richness (Pakhomov et al., 2006). Potential limitations of our analysis include differential vulnerability to fishing gear, which could be species-specific or relate to swimming performance or the size of individuals. The original experimental design attempted to cover all zones during the same season but some were sampled in separated cruises during summer and winter due logistical issues. Oceanographic variability may influence species distribution and potential seasonal changes of abundance and species richness need to be addressed in future research, especially at shallower depths (100–200 m). Also, the sampling effort was not evenly distributed throughout the whole of the latitudinal range with sites clustered within each zone; as such it is unlikely that all habitat types were sampled. This may be important as rocky substrates and other irregular habitats such as coral reefs and seamounts have been described as high diversity areas (hotspots), especially for cartilaginous fishes (Henry et al., 2013).

The clusters of sample sites also resulted in a relatively low resolution ‘picture’, and precluded a fine scale description of species’ distributions and abundance, and how these might be influenced by local conditions (e.g., habitat type).

Our results provide a quantitative description of species richness and abundance of the cartilaginous fish community on the outer continental shelf and slope of Chile to complement and extend knowledge of biological and ecological interactions of this demersal ecosystem. More than 90% of elasmobranch species worldwide inhabit demersal ecosystems on continental shelves and slopes (Compagno, 1990), which makes them vulnerable to trawl fishing (Shepherd & Myers, 2005) and we are just beginning to understand the potential ecological consequences of removal and declines of cartilaginous fishes. The information presented here is of immediate value in the assessment of the conservation status of species and the threats to their populations posed by demersal trawling. This study is also of particular value for future assessment of how natural or anthropogenic activities may impact the various species by providing quantitative baseline information against which change can be assessed.

The authors wish to thank all researchers involved during the project, especially to the staff of TecPes (Laboratorio de Tecnología Pesquera) at Pontificia Universidad Católica de Valparaíso (T Melo, CF Hurtado, D Queirolo, E Gaete, I Montenegro and R Escobar). Additional thanks to members of “Programa de Conservación de Tiburones, Chile” and ELASMOLAB staff at Universidad Austral de Chile for their valuable help with logistics, sampling and dissection help during fieldwork, especially J Lamilla, F Concha, H Flores, Y Concha-Perez and A Isla.

Additional Information and Declarations

Competing Interests

Author Contributions

Animal Ethics

Field Study Permissions

The authors declare there are no competing interests.

Carlos Bustamante conceived and designed the experiments, performed the experiments, analyzed the data, wrote the paper, prepared figures and/or tables.

Carolina Vargas-Caro performed the experiments, analyzed the data, reviewed drafts of the paper.

Michael B. Bennett analyzed the data, contributed reagents/materials/analysis tools, prepared figures and/or tables, reviewed drafts of the paper.

The following information was supplied relating to ethical approvals (i.e., approving body and any reference numbers):

This study was carried out in accordance with the “standards for the use of animals in research” approved by the Animal Care and Ethics Committee of the Universidad Austral de Chile (UACH/FIP 2005-61).

The following information was supplied relating to field study approvals (i.e., approving body and any reference numbers):

Capture of fishes during this study was permitted through Fisheries Undersecretariat Research Permit number 1959-06, 2931-06 and 181-07 issued by Ministry of Economy, Development and Tourism.

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
