# Peer review of "Biogeographic patterns in the cartilaginous fauna (Pisces: Elasmobranchii and Holocephali) in the southeast Pacific Ocean"

_PeerJ, doi:10.7717/peerj.416_

## Round 0.1 · original submission · Minor Revisions

· Academic Editor

Minor Revisions

Please respond positively to the suggestions of the reviewers. I would suggest that further analysis is not necessary in response to reviewer two (although you may want to confirm this) but some discussion of the existence of additional AND alternative clusters and classifications is warranted. I would also ask that you amend the abstract to highlight both congruence with existing work and the value of the work for non-specialist audiences. This is articulated in the discussion but not the abstract.

Reviewer 1 ·

Basic reporting

The article is well written, the figures and tables are clear, although some of them should be slighly modified.

Experimental design

This research took advantage of exploratory surveys designed with a different purpose. In this sense, the authors made a great job at extracting valuable information that otherwise will be discarded. A potential source of bias that is not discussed or considered in the analyses is that the southern areas were sampled during spring, whereas the northern areas were sampled in winter. In this way, potential seasonal effects may apply in addition to the latitudinal and bathymetric effects discussed in the paper. I understand that this is a byproduct of using data coming from sampling designs that are originally aimed at a different question, as is usual among many research projects on cartilaginous fishes. This should not prevent to publish the results (which are valuable), but the authors should address this potential source of bias, at least, as a paragraph in the Discussion.

Validity of the findings

Most patterns shown in the paper are clear. However, I think that an alternative cut value at a higher similarity level may provide a more informative community structure pattern. At present, Assemblage II includes 45 sites distributed along the whole study area. However, closer inspection of Fig. 5 reveals that this assemblage may be subdivided in clusters with a clear latitudinal signal. For example, at about 50% similarity, Assemblage II may be splitted in three clusters, one composed mostly of sites from areas 1, 2, 3 and 4, another cluster composed mostly of sites in areas 5 and 6 (plus just two from area 4 and one from area 3), and a third cluster composed of sites from areas 7 and 8. The remaining assemblages are mostly unchanged except for assemblages I and III, which are further subdivided into 3 and 2 clusters, respectively. This scheme may be of better resolution than the used presently. It is important that the meaning of the codes of each sample in the cluster of Fig. 5 (and 6) be explicitly explained in the figure legend (i.e. Does the first number refer to each area? What does the letter mean? And the final number?).

Additional comments

Minor comments:

Line 57: A verb (e.g. give, offer) is lacking in this sentence between “studies” and “a”: “While fishery-dependent studies a description of diversity...”.

Line 72: What does PAM mean?

Line 77: Please delete “average”; a range is given rather than an average.

Lines 83-85: The maximum depth of tows is given as 499 m, however in Fig. 1, several tows appear beyond the 500 m isobath. In a region with such a steep slope as Chile, this may be well deeper than 500 m. Please check.

Line 102: Please define Frequency of occurrence.

Line 126: “sadowskii” instead of “sadowsky”. Although Torpedo tremens is usually referred to as a ray, Rajella sadowskii belongs in the family Rajidae, so the term skate is commonly applied to it. This would make unnecesary the inclusion of R. sadowskii in this sentence.

Line 129: Delete “with”.

Table 2: This is actually two tables combined into one. I would recommend to split this table into two ones.

Lines 135-138: I cannot fully understand this sentence. Fig. 2 shows several regression lines, which regressions is this sentence referring to?

Line 169: “griseus” instead of “griseous”.

Lines 180, 190, 211, 317: “Bythaelurus” instead of “Bythalelurus”.

Line 282: Fig. 7 does not show depth ranges.

Lines 332-345: An additional hypothesis that the authors should consider to explain the absence of large sharks is that samples were taken exclusively by trawling. Large sharks, especially lamnoids and carcharhinoids, appear in trawl samples very rarely. My personal experience is that, in the same area, trawl sampling will not catch a single large shark, while at the same time, sampling with hook gears will catch them. I suggest to add a comment on this potential sampling bias in this paragraph.

Lines 355-368: This kind of limitations are the ones that may potentially affect the sampling of large sharks. Some of this should be said in lines 332-345.

Line 381: Please spell out “TecPes PUCV
”.

Line 387: Delete “had”.

·

Basic reporting

The manuscript “Biogeographic pattern in the cartilaginous fauna (Pisces: Elasmobranchii and Holocephali) in the southeast Pacific Ocean” by Bustamante et al. describes the assemblages of sharks, skates and chimaeras off the coast of Chile through fishery independent trawl surveys. The paper is well written, generally clear and free from excessive grammatical and spelling errors. Sufficient background and introductory material is provided to place the study into context and the methods and materials are well described.

A detailed list of edits and annotations is provided as an attachment. Please read these over and clear up any confusion. Although the text is generally well written there are still errors that should be addressed. The one recurrent error is the overuse of the semicolon. Many times in the text, a semicolon is used when a comma was appropriate. Please revise all the uses of the semicolon in the manuscript and ensure that it is appropriate. You can refer to materials like this (https://owl.english.purdue.edu/owl/resource/607/04/) for help.

The figures and tables are clear and informative. There are no redundancies or irrelevant figures. The legends of figures 5 and 6 could use a line about the meaning of the site code (e.g. 7A.2, 5B.4, etc…). This will help the reader pick up patterns. Figure 1 is good, and if it is possible to add the areas of high fishery CPUE, that might be even more useful.

Experimental design

The manuscript clearly outlines a research question and the methodology is relevant for addressing that question. Overall, the experimental design, fieldwork and analyses are appropriate. There are a few small areas that are not entirely clear, and hopefully my annotations will help clear those up. Also, as noted above, the site codes should be explained in the main text of the manuscript.

Although it is well understood that survey data is not perfect and that the available data should be used, there are some flaws in the data collection that should be acknowledged by the authors. The primary one is the difference of timing in the tows at different sites. As mentioned on lines 85-87, all sites were not towed in the same season and there is a bias as to the timing of sampling of different latitudes. Since latitudinal gradients are discussed quite heavily in the paper, there should also be a discussion as to the potential effects of the sampling timing at different sites. This would work quite well in the last section of the Discussion.

Validity of the findings

The results and interpretation of the results are appropriate, relevant and statistically sound, and provide a significant addition to the knowledge base of fish community assemblages. The authors do a good job of providing a review of existing data and how this study contrasts and adds to the existing knowledge base.

Additional comments

I recommend this manuscript be accepted for publication after the edits and suggestions made here are incorporated or addressed. These will help with the clarity and thoroughness of the manuscript.

---

## Round 0.2 · accepted · Accept

· Academic Editor

Accept

Thank you for constructively addressing the reviewers concerns.